# Effect of 3D Printing Process Parameters on Damping Characteristic of Cantilever Beams Fabricated Using Material Extrusion

**DOI:** 10.3390/polym15020257

**Published:** 2023-01-04

**Authors:** Feiyang He, Haoran Ning, Muhammad Khan

**Affiliations:** 1Centre for Life-Cycle Engineering and Management, Cranfield University, Cranfield MK43 0AL, UK; 2School of Aerospace, Transport and Manufacturing, Cranfield University, Cranfield MK43 0AL, UK

**Keywords:** material extrusion, damping, nozzle size, infill density, pattern

## Abstract

The present paper aims to investigate the process parameters and damping behaviour of the acrylonitrile butadiene styrene (ABS) cantilever beam manufactured using material extrusion (MEX). The research outcome could guide the manufacture of MEX structures to suit specific operating scenarios such as energy absorption and artificially controlled vibration responses. Our research used an experimental approach to examine the interdependencies between process parameters (nozzle size, infill density and pattern) and the damping behaviour (first-order modal damping ratio and loss factor). The impact test was carried out to obtain the damping ratio from the accelerometer. A dynamic mechanical analysis was performed for the loss factor measurement. The paper used statistical analysis to reveal significant dependencies between the process parameters and the damping behaviour. The regression models were also utilised to evaluate the mentioned statistical findings. The multiple third-order polynomials were developed to represent the relation between process parameters and modal damping ratio using stiffness as the mediation variable. The obtained results showed that the infill density affected the damping behaviour significantly. Higher infill density yielded a lower damping ratio. Nozzle size also showed a notable effect on damping. A high damping ratio was observed at a significantly low value of nozzle size. The results were confirmed using the theoretical analysis based on the underlying causes due to porosity in the MEX structure.

## 1. Introduction

Additive manufacturing (AM), referring to 3D printing, was developed in the late 1980s. After nearly 40 years of development, due to several advantages such as fast prototype, high level of customisation and cost-effectiveness, this advanced manufacturing technology has been used in a wide range of industries, including aerospace [1], medical [2], biomanufacturing [3], etc. Of course, fused deposition modelling (FDM), as a material extrusion (MEX) technology defined using ISO/ASTM 52900 [4], is also included in these applications. It uses a heated nozzle to melt raw material filament to a semi-liquid state and extrudes material from the nozzle layer by layer to print the whole structure [5].

The rapid development of FDM technology and its numerous applications has naturally attracted academic research attention. A large number of studies investigated the mechanical properties and behaviours of FDM structures [6,7]. In contrast to structures produced using conventional manufacturing techniques, the mechanical properties of FDM parts depend not only on the raw material itself but are also significantly influenced by the process parameters of the printing process. The parameters can be categorised as (1) manufacturing parameters, such as nozzle temperature and printing speed, and (2) structural parameters, including infill density, raster orientation, nozzle size, and layer thickness [8]. Despite the diversity and complexity of these parameters, the relationship between them and their properties has been studied extensively.

Several studies investigated the effect of process parameters on static mechanical properties such as tensile modulus, tensile strength and fracture toughness [9,10,11,12,13,14,15,16,17,18,19]. On the other hand, some papers focused on fatigue behaviour, such as fatigue life and strength [5,16,20,21,22,23,24,25,26,27,28]. However, there is no in-depth understanding of the process parameter’s influence on damping behaviour in FDM structures.

Damping causes energy dissipation during mechanical oscillations. Especially for the various thermoplastic materials commonly used as raw materials for FDM, damping becomes a significant mechanical property due to the viscoelastic material behaviour. Similar to parts produced using conventional manufacturing techniques, raw material types can affect the structural damping of FDM parts. Ge et al. compared the damping behaviour of 3D-printed Tango Black Plus resin with traditional cushioning materials, such as polyethylene, under impact loading. The resin showed extreme damping and was capable of absorbing almost 100% of the impact energy and recovering fully to the original dimensions after multiple platen drop tests [29]. Zhang et al. developed a Magnesium-Nitinol (Mg-NiTi) composite using a selective laser melting technique via the infiltration of magnesium melt into a 3D-printed Nitinol scaffold. The 3D-printed Mg-NiTi exhibits great damping capacities and an exceptional energy absorption efficiency [30]. Similarly, Wang et al. proposed a viscoelastic material filling (VMF) method to balance structural and vibrational performance. Through selective laser sintering technology, they filled the 3D-printed Kagome lattice with viscoelastic thermosetting polyurethane. The filling material provided high stiffness and considerable damping performance [31].

In addition to raw materials, 3D printing parameters have a more significant impact on damping properties. Structural infill properties are one of the critical parameters. León-Calero et al. employed compression tests to investigate the specific energy absorption (SEA) and specific damping capacity (SDC) of the 3D-printed thermoplastic polyurethanes. Their research established the relationship between the geometrical characteristics (infill density and type of infill pattern) and the damping behaviours of the printed part. The paper concluded that optimal SEA and SDC performances were obtained for a honeycomb pattern at a 50% infill density [32]. Moreover, Francisco Medel et al. assessed the damping ratios for 3D-printed polylactic acid (PLA) beam produced with various parameters (build orientation, nozzle temperature, layer height, print speed and raster angle). The research confirmed the association between high damping and poor inter-filament bonding [33].

The above review shows that only a few studies discussed the effect of 3D printing parameters on damping properties, while most of the studies just introduced the process parameters’ influence on the common static and dynamic mechanical properties. The published research on process parameters and damping behaviour is limited to experimental results and observations without a distinct mathematical model which can be used to analyse their dependencies. Therefore, this paper presents an effort to develop a mathematical model to relate the dependencies of process parameters (nozzle size, infill density and infill pattern) on the damping ratio. The paper used analysis of variance (ANOVA) and correlation analysis to reveal the statistically significant dependencies between parameters and damping properties. The regression models were also utilised to evaluate the mentioned statistical findings. The multiple third-order polynomials were developed to represent the relation between the process parameters and the modal damping ratio using stiffness as the mediation variable. The obtained results showed that the infill density affected the damping behaviour significantly. Higher infill density yielded a lower damping ratio. Nozzle size also showed a notable effect on damping. A high damping ratio was observed at a significantly low value of nozzle size. The results were also confirmed using the theoretical analysis based on the underlying causes due to porosity in the fused deposition modelling (FDM) structure.

## 2. Methodology

### 2.1. Identification of Damping Properties

Damping is generally derived from internal friction during material deformation [34]. There are numerous approaches to describe damping mathematically [35]. One of them is the viscous damping theory [36], as shown in Equation (1).
(1)$$F_{D}\left(t\right)=c\dot{u}\left(t\right)$$
where the scale *D* in $F_{D}\left(t\right)$ denotes damping; $F_{D}\left(t\right)$ is the damping force; c is the damping coefficient; and $\dot{u}\left(t\right)$ is the velocity of motion. In practice, it is common to use the modal damping ratio ζ to represent the damping coefficient c due to their linear relationship.

Another well-known damping theory is hysteresis damping, or structural damping, which is more commonly used due to its better representation of the energy dissipation mechanism of internal friction in materials [35]. It can be expressed in complex form Equation (2).
(2)$$F_{D}\left(t\right)=i\eta ku\left(t\right)$$
where i is the imaginary number, η is the loss factor; k is the stiffness. The loss factor η can be defined in Equation (3)
(3)$$\tan\delta =\frac{E^{\prime \prime }}{E^{\prime }}=\eta$$
where δ is the phase shift between stress and strain; $E^{\prime \prime }$ is the loss modulus; and $E^{\prime }$ is the storage modulus. Due to the different behaviours between the two damping expressions, this paper investigated both damping properties using modal damping ratio and loss factor, respectively.

### 2.2. Specimen Fabrication

As one of the most widely used FDM thermoplastic materials, ABS was employed for all specimens in the research, similar to our previous studies [37,38,39,40,41,42,43]. Some raw material specifications are listed in Table 1. The specimen was designed as a 2 mm thick cantilever beam with the geometry shown in Figure 1. The cantilever beam provides excellent free vibration response under the impact force and facilitates subsequent data processing.

The CAD model of the specimen was developed in SolidWorks^©^ and imported into Ultimaker Cura^©^ for setting the process parameters and slicing. Most print parameters were set using the software default values (layer height: 0.06 mm, wall thickness: 1.05 mm, nozzle travel speed: 120 mm/s, print speed: 45 mm/s, nozzle temperature: 240 °C, bed temperature: 90 °C) to seek a balance between printing quality and manufacturing time. Later, the sliced model was exported to G-CODE files and sent to Ultimaker 2+ for printing.

### 2.3. Design of Experiment

#### 2.3.1. Process Parameter Determination

It is critical first to determine the parameters investigated in the research. Previous studies analysed most process parameters’ relationships with common mechanical properties. Following the thought of damping change due to different 3D-printed particle sizes [44], the nozzle size, determining the filament diameter in FDM structures, can similarly affect the damping properties. Therefore, nozzle size was investigated in our research with three levels: 0.4, 0.6 and 0.8 mm.

On the other hand, as several studies proposed that infill density and infill pattern can significantly affect the strength and stiffness of FDM structure [32,45,46,47], these two parameters were also tested. Four levels (40, 60, 80 and 100%) and types (line, gyroid, cubic and triangles) were considered for infill density and infill pattern, respectively. It was worth noting that the three different raster angles (0, ±45 and 90°) were further subdivided in the line print pattern. Therefore, the research used a total of 72 process parameter combinations, as shown in Figure 2. For each combination, three specimens were printed and tested to ensure the repeatability of the experiments.

#### 2.3.2. Experimental Setup and Procedures

The specimen, as a cantilever beam, has infinite freedom. It means that there is infinite damping coefficient $C_{j}$ and damping ratio $\xi_{j}$ for $j^{th}$ mode, respectively. This research only measured the modal damping ratio for the first mode using impact test.

The experimental setup for the impact test is shown in Figure 3. The specimen was fixed on a test rig. A full constraint condition was applied on the beam’s fixed end. One accelerometer (PCB 352A21 model, PCB Piezotronics, Depew, NY, USA) was fixed on the beam’s free end, which can monitor the acceleration values in real-time. A slight impact force was applied to the beam during the test. Meanwhile, the DAQ card (NI 9234) and DAQ chassis (NI 9174) (National Instrument, London, UK) transferred the data from the accelerometer to the computer. The Signal Express^©^ 2015 saved the data as a .txt file which can be imported into MATLAB R2021a for the following process. Three damping ratios were collected for each process parameter combination using testing three identical specimens.

The DMA was conducted using DMA Q800 manufactured by TA Instruments (New Castle, DE, US) to measure the loss factor after the impact test. The specimen was clamped at both ends, and one end was flexed, as shown in Figure 4. Sinusoidal stress was applied to the specimen with a 1 Hz frequency to ensure a 1 mm displacement amplitude, and the corresponding strain was measured. The phase difference δ between the two sine waves and the stiffness of the specimen were measured. A total of 24 phase shift values were collected for each parameter combination for the following analysis.

### 2.4. Experimental Data Post-Process and Analysis

#### 2.4.1. Damping Ratio Calculation

The data from the accelerometer recorded the times and corresponding acceleration amplitudes. Therefore, the modal damping ratio can be calculated using the logarithmic decrement δ, which is the logarithm of the ratio between the amplitudes of two subsequent peaks, $u_{i}$ and $u_{i+1}$, as shown in Equations (4) and (5),
(4)$$\delta =\ln\frac{u_{i}}{u_{i+1}}=\ln\frac{{\ddot{u}}_{i}}{{\ddot{u}}_{i+1}}=\frac{2\pi \xi }{\sqrt{1-\xi^{2}}}$$
(5)$$\xi =\frac{\delta }{\sqrt{{\left(2\pi \right)}^{2}+\delta^{2}}}$$

However, due to the small damping of the system, the free vibration decayed slowly, and in this study, the damping ratio ξ was calculated using the amplitude ratio of accelerations, ${\ddot{u}}_{n}$ and ${\ddot{u}}_{n+m}$, between m=5 cycles to obtain higher accuracy, as shown in Equations (6) and (7),
(6)$$\delta =\ln\frac{{\ddot{u}}_{n}}{{\ddot{u}}_{n+m}}$$
(7)$$\xi =\frac{\delta }{\sqrt{{\left(2\pi m\right)}^{2}+\delta^{2}}}$$

#### 2.4.2. Statistical Analysis

The collected experimental data were then analysed using several statistical methods for qualitative and quantitative assessment of the effect of different process parameters on modal damping and loss factor, respectively.

Analysis of variance (ANOVA) was used to perform a preliminary evaluation of the experimental data [48]. The research assessed whether the total variance in the test results was a combination of differences due to random factors (i.e., experimental error) and different levels of the process parameters. A print parameter was considered to have a significant effect (level of significance *p* < 0.05) on the damping behaviour if the variance arising from the different levels of process parameters was much larger than those caused by the random factor; otherwise, it is considered to have no significant effect.

The ANOVA results provided an initial qualitative indication of whether each print parameter and their interactions influenced the damping behaviour. Later, the correlation analysis was performed to compare the linear correlation between each process parameter and damping properties. Because the infill pattern parameter was a categorical variable and difficult to quantify numerically, we calculated the Pearson correlation coefficient r, which was used most, between quantitative parameters (nozzle size and infill density) and damping properties. The results of r quantified the level of significance of the effect of different process parameters on structural damping performance.

Furthermore, this research developed regression models for quantitative assessment of the effect of nozzle size and infill density. In particular, the interaction between them was investigated with moderator analysis using a standardised linear regression model. Later, the appropriate polynomial was fitted between process parameters and modal damping factors using a mediation variable, i.e., structural stiffness.

## 3. Results and Discussion

### 3.1. Effect of Process Parameters on Modal Damping Ratios

The ANOVA results of different process parameters are demonstrated in Table 2. All three print parameters had extremely small *p*-values, which confirmed that they significantly affected the modal damping ratio in the tests. The infill pattern was the most significant process parameter, followed by the infill density. The nozzle size was considered to have the least significant effect on the modal damping ratio as it had the highest *p*-value.

Figure 5, Figure 6 and Figure 7 show the box plots for each process parameter. Combined with the correlation coefficient analysis results, shown in Table 3, trends in the modal damping ratio were analysed further. Although ANOVA’s results in Table 2 suggested that the infill pattern was the most critical parameter, the range of mean modal damping ratios was only between around 0.017 and 0.024 for the different patterns. Figure 5 also did not reflect much change between each pattern, and they were relatively similar. However, in the single line pattern subset, the damping ratio possessed a significant trend for various raster angles. The 0° raster angle resulted in the lowest damping ratio value, whilst the 0.021, 45° raster angle yielded a higher mean of ξ, 0.022. The highest mean, ξ, 0.024, was attained with a 90° raster angle. The box plot in Figure 5 also demonstrated the same trend, where the higher raster angle increased the modal damping.

Regarding infill density, both Table 3 and Figure 6 demonstrated that it had a negative impact on the modal damping ratio. Thus, mean ξ significantly reduced from 0.025 to 0.016 when the infill density increased from 40% to 100% in Table 2. Likewise, we were able to detect a macro downward trend for all modal damping ratio data with an increased infill density in the boxplot Figure 6. This argument can also be confirmed by the negative r value, −0.406, in Table 3.

Nozzle size exhibited a lower significance as its higher *p*-value in ANOVA compared to the infill density. The specimens printed with the 0.4 mm nozzle were associated with a significantly higher mean ξ (0.025). The 0.6 mm nozzle sizes led to a comparatively lower mean ξ (0.019). Meanwhile, the 0.8 mm nozzle size produced a relatively lower mean ξ (0.018). Similarly, the boxplot in Figure 7 showed a decreased trend, and data scatter distribution shifted to a lower position when the nozzle size increased. The correlation analysis between the nozzle size and modal damping ratio reported a −0.270 r value, which also supported the inverse relationship between them. In addition, it is worth mentioning that the correlation coefficient of the nozzle size was lower than the infill density in terms of absolute value. Thus, it confirmed that the infill density demonstrated higher significant effects than the nozzle size, corroborating ANOVA’s results.

### 3.2. Effect of Process Parameters on Loss Factor

Similar to the statistical analysis in Section 3.1, ANOVA and correlation analysis were carried out to determine the potential relationships between process parameters and loss factor. Nevertheless, the *p*-values of all process parameters in ANOVA exceed the 0.05 level of significance (see Table 4), suggesting that no significant correlation between the process parameters and the loss factor exists.

Likewise, the correlation analysis results also demonstrated a high *p*-value, as shown in Table 5, which confirms the insignificance of the process parameters on the loss factor. This result is reasonable because the loss factor is one mechanical property that essentially depends on the material itself. The microstructure change due to various process parameters cannot affect the loss factors in this research because the material was the same in all tests. This conclusion is aligned with Equation (3), too. Since the loss factor η is a dimensionless value as a ratio that is between the loss modulus and storage modulus, it eliminates the influence of the structure and is only affected by the material itself.

### 3.3. Regression Model Development

Section 3.1 investigated the effect of each process parameter individually on the damping behaviour. The interaction between these parameters was further assessed with the moderator analysis using a standardised linear regression model detailed in this section. The model determines whether the relationship between one process parameter and the modal damping ratio depends on the value of the other process parameters.

In order to obtain the model, the experimental data were fitted with the linear regression form as shown in Equation (8) (it is worth mentioning that only the nozzle size and infill density were considered because the pattern parameter was difficult to quantify as a categorical variable):(8)$$\xi =p_{0}+p_{1}x_{1}+p_{2}x_{2}+p_{3}x_{1}x_{2}$$
where p is the regression coefficient, $x_{1}$ and $x_{2}$ represent nozzle size and infill density, respectively. The interaction between these two parameters can be determined by evaluating the significance of $p_{3}$. Table 6 shows the fitted linear regression coefficients and statistical analysis. Interestingly, the p value for $p_{3}$ is 0.126, which exceeds the level of significance of 0.05. Therefore, it confirms that there is no significant interaction between the two process parameters.

After ruling out potential interactions, regression models were developed between the process parameters and the modal damping ratio. However, none of the results were satisfactory. The research had to find a potential mediation variable to build their connections. Consequently, the relationship between them and the stiffness was utilised.

Several studies have proposed that structural stiffness was strongly correlated with damping [49]. Previous research also confirmed a negative correlation between stiffness and internal viscous damping [50]. Thus, this paper selected stiffness as a mediation variable and investigated their dependencies accordingly. The stiffness values of the specimens were obtained from the DMA result. The linear regression model was developed between the stiffness and damping ratio. The correlation curve with R-square 0.24 is depicted in Figure 8. It suggested that a higher stiffness might be associated with a relatively low damping ratio, which will be confirmed with the underlying mathematical dependencies of Equation (18), derived in Section 3.4. The equation also reveals the reasons for the low R-square value of the regression model. It is due to the nonlinear relationships between the stiffness and damping ratio and the combined effect of the distributed mass variable.

After that, a suitable nonlinear third-order polynomial was fitted between process parameters and stiffness (unit: N/m) for the experimental data with all patterns. The poly (2,3) model was developed with the form shown in Equation (9) and regression coefficient values in Table 7. Where P is the regression coefficient, x denotes the nozzle size (unit: mm) and y denotes the infill density (%). The fitting surface is presented in Figure 9 with an R-square value of 0.75, which is relatively high.
(9)$$stiffness=P_{00}+P_{10}x+P_{01}y+P_{20}x^{2}+P_{11}xy+P_{02}y^{2}+P_{21}x^{2}y+P_{12}xy^{2}+P_{03}y^{3}$$

It is worth noting that the scatters employed in regression incorporated all raw data from the specimens with all pattern parameters. It implies that this model does not consider the effect of the pattern. Therefore, the enhanced regression models for specimens with different patterns were proposed further with the idea that the constant term $P_{00}$ in Equation (9) was as a pattern-dependent coefficient. Meanwhile, other regression coefficients in Table 7 were kept constant. Thus, the regression surface in Figure 9 could offset along the z-axis to fit different pattern data. The different pattern-dependent coefficients and corresponding regression R-square values are listed in Table 8. It demonstrated the extremely high R-square values (0.88–0.97) for all MLR models, which means the pattern parameter appears to have a linear relationship with stiffness.

Figure 10 shows the shifted fitting surface for all six patterns. It exhibits that infill density significantly affects stiffness. In comparison, nozzle size appears to have a slight influence on stiffness. The slightly higher infill density leads to a rapidly increasing trend of stiffness. Meanwhile, the change in stiffness due to the nozzle size looks insignificant. The reason is that stiffness is linearly related to the square of nozzle size and the cubic of infill density in Equation (9). This observation is in agreement with the effect of both process parameters on damping ratios and confirms the argument in Section 3.1. However, it is worth noting that the mathematical model proposed in this section is not universal. Because the damping ratio is highly dependent on the structural geometry and dimensions, the proposed mathematical relationship can only provide the trend instead of the exact values.

### 3.4. The Theoretical Justification for Process Parameter Influence on Damping Behaviour

The statistical analysis results in Section 3.1 demonstrated that all process parameters in the tests had a significant influence on the modal damping ratio. To understand the underlying causes, this section explored the nature of damping, incorporating cantilever beam vibration and the material properties of FDM structures.

The modal damping ratio is used in this paper to represent the damping behaviour of the structure. It is defined as the ratio between the damping coefficient c and the critical damping coefficient $c_{cr}$, denoted by Equation (10) as the most straightforward single-degree-of-freedom system.
(10)$$\xi =\frac{c}{c_{cr}}$$
where the system critical damping coefficient is
(11)$$c_{cr}=2\sqrt{km}$$
where k is system stiffness and m is mass. Thus, the damping ratio ξ is actually determined by structural damping, stiffness and mass together. However, the critical point to remember is that the damping here is a theoretical parameter representing the energy dissipation, and it is not an actual physical parameter like stiffness and mass. A change in these three parameters will ultimately result in a different damping ratio, which is the fundamental reason for the change in the structural damping ratio.

The cantilever beam tested in this study is not a single-degree-of-freedom spring-mass-damping vibration system. Whereas the principle of damping ratio change is the same as that of a simple system, this research ignored the air’s damping force on the beam’s movement and considered mainly the damping stresses due to the repeated deformation of the fibres in the beam’s cross-section, where the mechanical energy is dissipated into heat. The damping stress σ is generated with the distribution along the cross-section height and can be expressed with Equation (12):(12)$$\sigma =c_{s}\frac{\partial \varepsilon }{\partial t}$$
where $c_{s}$ is the equivalent viscous damping coefficient for viscoelastic beams, ε is the bending strain along the beam cross-section and t is the time. The equivalent critical damping coefficient $C_{cr,j}$ and damping coefficient $C_{j}$ for $j^{th}$ mode are written as Equations (13) and (14), respectively.
(13)$$C_{cr,j}=2\omega_{j}M_{j}$$
(14)$$C_{j}=\frac{c_{s}\omega_{j}^{2}}{E}M_{j}$$
where $\omega_{j}$ is the natural frequency for $j^{th}$ mode as shown in Equation (15) and $M_{j}$ represents the generalised mass for $j^{th}$ mode as shown in Equation (16):(15)$$\omega_{j}=\alpha_{j}^{2}\sqrt{\frac{EI}{\rho AL^{4}}}$$
(16)$$M_{j}=\int_{0}^{L}\rho AY_{j}^{2}\left(x\right)dx$$
where $\alpha_{j}$ is the modal constant, E is the tensile modulus, I is the moment of inertia for the beam’s cross-section, ρ is the structural density, A is the cross-section area, L is the beam length and $Y_{j}$ is the beam’s $j^{th}$ mode shape. Hence, we derived the generalised damping coefficient $C_{j}$ and modal damping ratio $\xi_{j}$, as shown in Equations (17) and (18):(17)$$C_{j}=\frac{\alpha_{j}^{4}c_{s}I}{L^{4}}\int_{0}^{L}Y_{j}^{2}\left(x\right)dx$$
(18)$$\xi_{j}=\frac{\alpha_{j}^{2}I}{2L^{2}}c_{s}\sqrt{\frac{1}{\rho AEI}}$$

The $\xi_{1}$ in Equation (18) is the modal damping ratio actually measured in the experiments. Hence it is proportional to the term $\left(c_{s}\sqrt{\frac{1}{\rho AEI}}\right)$ which includes the damping coefficient $c_{s}$, distributed mass (ρA) and distributed stiffness (EI). Therefore, the process parameters influenced the damping behaviour by essentially affecting these structural properties.

Although $c_{s}$ in Equation (18) is a theoretical representation of the complex energy dissipation, it is numerically associated with several practical damping sources from a physical point of view. This can be partly attributed to the energy dissipation in ABS fibre due to the sliding of molecular chains as a kind of internal friction [51,52], which is the same for all specimens in the experiments. On the other hand, the porosity in the FDM micro-structure also leads to another significant influence. The effect of nozzle size on damping might stem from this feature. Within FDM structures inherently exist air voids, as shown in Figure 11a and captured using a scanning electron microscope (SEM). The black, evenly spaced areas clearly represent the air voids between the ABS filaments. These air voids in the FDM structure are like miniature joints embedded in the structure. This porosity suggests poor interfilamentous bonding in the void area. Therefore, the joined filament surfaces are easier to slide relative to each other during the vibration. Eventually, it results in energy dissipation and affects $c_{s}$. It is worth mentioning that the nozzle size has a notable effect on the porosity percentage in the FDM structure. Figure 11b,c, captured using the DinoLite digital microscope, exhibit the beam cross-section printed with different nozzle sizes. Interestingly, the beam’s cross-section, printed using a 0.8 mm nozzle, attains the lowest air void density, implying that it has the least micro-joints per unit area. Thus, the energy dissipation due to interfilamentous surface friction is less than the other two smaller nozzles. This leads to a lower $c_{s}$ value. Also, because the nozzle size has a positive influence on the distribution stiffness [37], the 0.8 mm nozzle size yields the lowest modal damping ratio ξ. Conversely, the reduced nozzle size increases the number of filaments per layer and is associated with more air voids in the structure. It results in a higher ξ value.

The author’s previous studies [37] have confirmed that the alignment between the filament and load direction significantly affects the anisotropic material properties for the raster angle influence in the line pattern parameter. A 90° raster angle, perpendicular to the test’s bending stress, provides the weakest bonding between the filament surfaces along the loading direction. Therefore, the slides between filaments become easier during vibration. This implies more mechanical energy loss leading to a higher $c_{s}$ value and eventually yielding a higher damping ratio. Conversely, the lowest damping ratio was attained when printing with a 0° raster angle.

Both the nozzle size and raster angle influence the modal damping ratio through their effect on the equivalent viscous damping coefficient $c_{s}$. Nevertheless, the infill density, demonstrating a more significant impact, acts on ξ in a different way. Several specimens with different infill densities are presented in Figure 12. It visually shows the changes in the internal gaps between skeleton construction ranging from a 40% to 100% infill density. Apparently, it has a critical influence on structural distributed mass and stiffness. A lower infill density significantly decreases the distributed mass and stiffness in Equation (18). Thus, the corresponding structure attains a higher modal damping ratio.

### 3.5. Comparison with the Previous Related Studies

As stated in the introduction section, a large number of studies have investigated the influence of process parameters on the mechanical properties of MEX structures. Table 9 summarises the works mentioning the infill density, nozzle size and infill pattern.

Most of them followed the ASTM standards to measure the MEX structure’s tensile strength and concluded that a higher infill density yielded a higher strength. Meanwhile, only two studies [32,33] investigated the damping behaviours of the MEX structure. Only one study [33] tested the influence of the raster angle on the modal damping ratios. Therefore, very few previous methods and conclusions are available for comparison.

This study carried out the impact test for specimens with a cantilever beam structure, which was the same as the method in [33]. However, interestingly, the modal damping ratio exhibited an opposite tendency with different raster angles when comparing our results with [33]. This study suggested a reduced value with an increased raster angle, but an increasing trend was observed when increasing the raster angle in [33]. Despite the fact we provided the theoretical justification in Section 3.4 and that there was no explanation in [33], this opposite trend still needs further investigation.

## 4. Conclusions

The present study was designed to determine the effect of process parameters (nozzle size, infill density and pattern) on the damping properties (damping ratio and loss factor) for the FDM ABS cantilever beam. It appears to be the first study to assess the FDM structural damping behaviours with these process parameters.

A mathematical model was developed to introduce the dependencies between the modal damping ratios and structural stiffness. The current data suggest that the cantilever beam with a higher stiffness exhibits a lower damping ratio. Moreover, this study proposed an empirical model for the relationships between the process parameters and stiffness, indicating that stiffness is linearly related to the square of nozzle size and the cubic of infill density. The present models have been the first attempt to establish quantitative interdependencies between process parameters, stiffness and modal damping ratios.

Furthermore, the research outcomes demonstrate that the infill density has the most significant impact on damping behaviour, and a higher infill density yields a lower damping ratio. Nozzle size also has a notable effect on damping, and the damping ratio reached the highest value when using the smallest nozzle. The results were also confirmed by using the theoretical analysis based on the underlying causes due to the porosity in the fused deposition modelling (FDM) structure. This is the only empirical and theoretical investigation into the impact of process parameters, and it has gone some way towards enhancing our understanding of the damping behaviour of the 3D-printed structure.

## Figures and Tables

**Figure 1 polymers-15-00257-f001:**
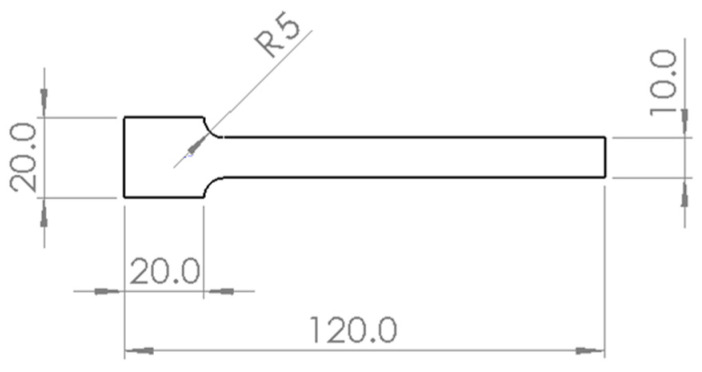
Geometrical dimension of samples (unit: mm).

**Figure 2 polymers-15-00257-f002:**
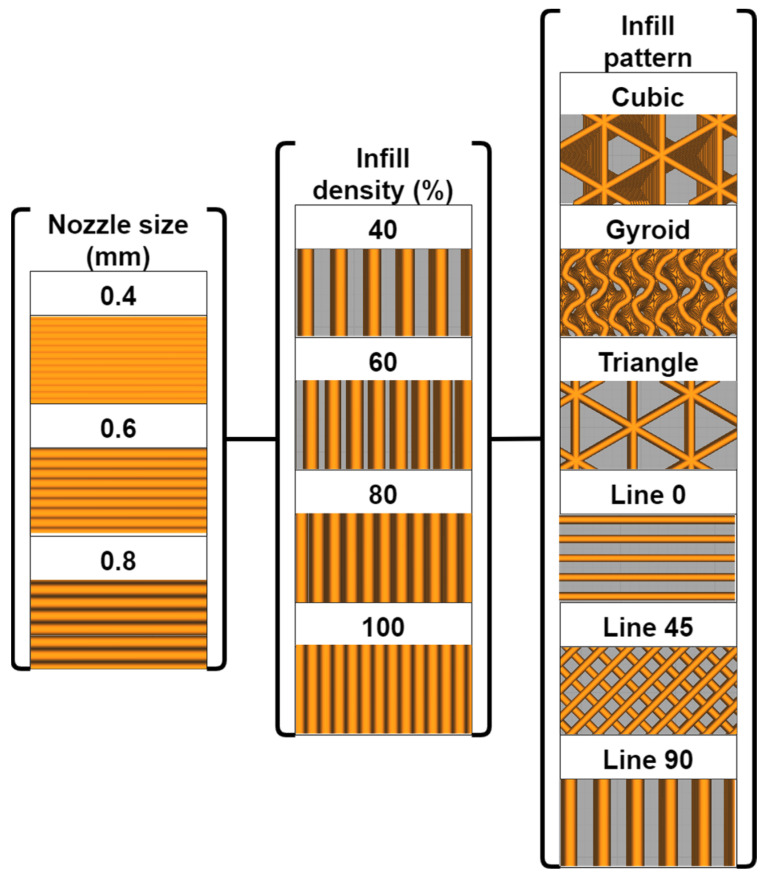
Process parameter levels and schematic.

**Figure 3 polymers-15-00257-f003:**
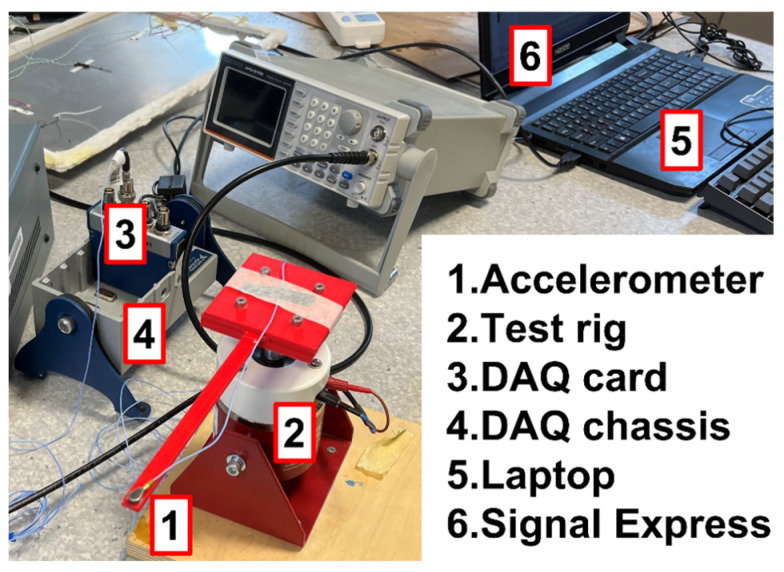
Experimental setups.

**Figure 4 polymers-15-00257-f004:**
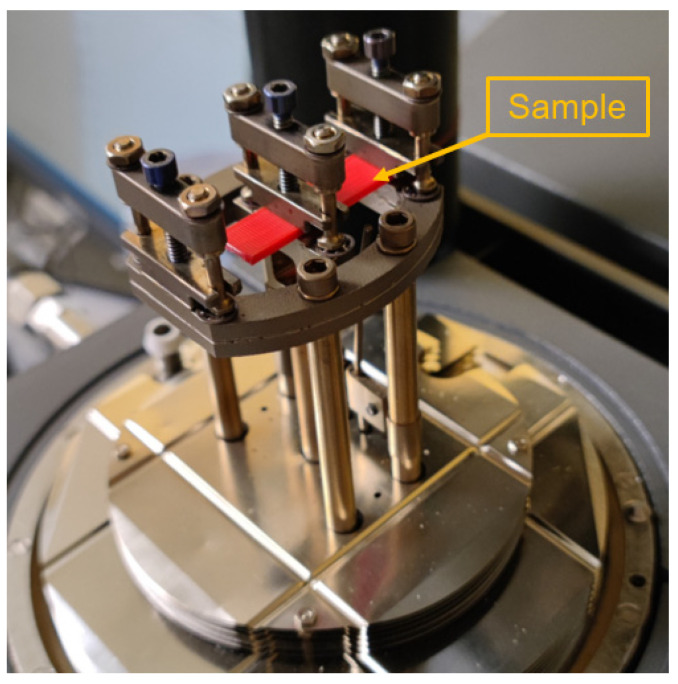
DMA test setup.

**Figure 5 polymers-15-00257-f005:**
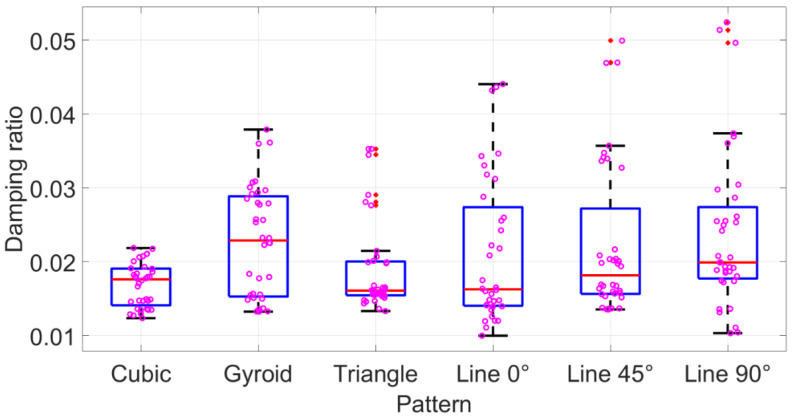
Damping factor box plots for infill pattern. Red dots are outliers. Violet circles are raw data.

**Figure 6 polymers-15-00257-f006:**
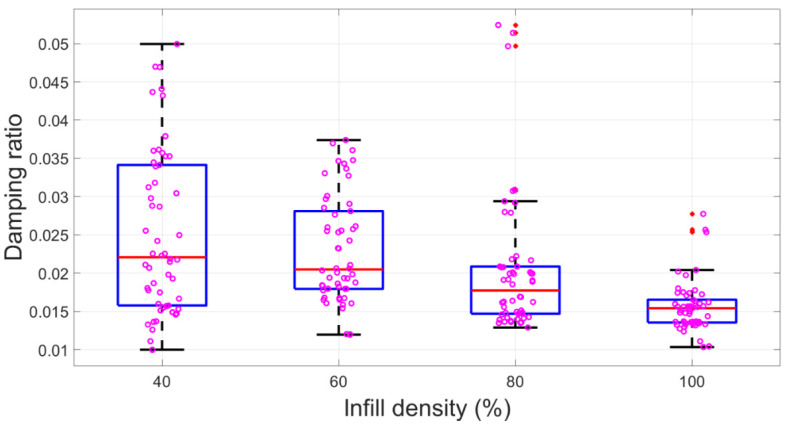
Damping factor box plots for infill density. Red dots are outliers. Violet circles are raw data.

**Figure 7 polymers-15-00257-f007:**
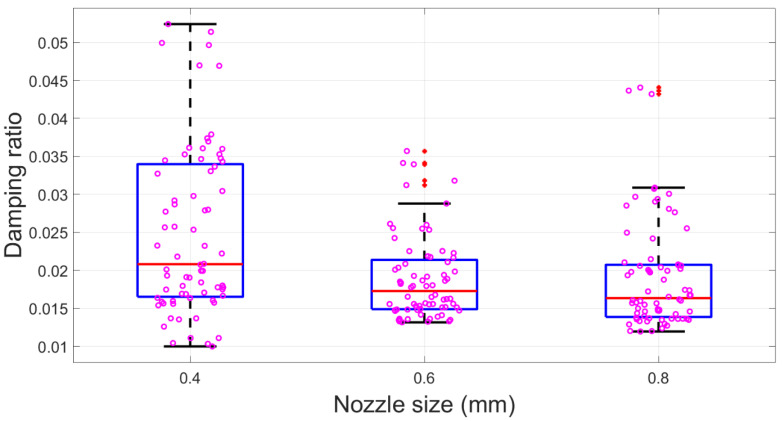
Damping factor box plots for nozzle size. Red dots are outliers. Violet circles are raw data.

**Figure 8 polymers-15-00257-f008:**
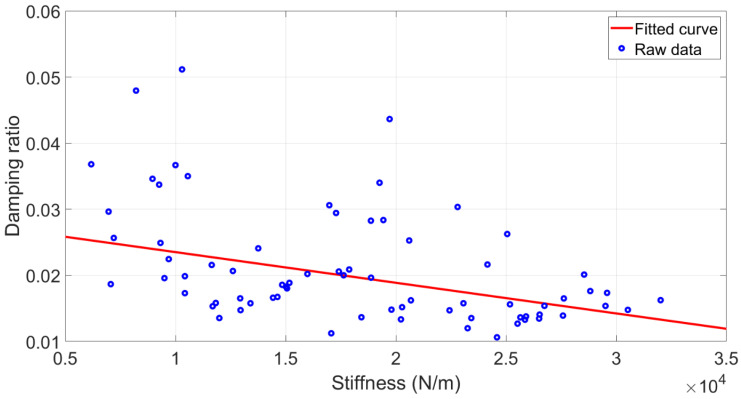
The fitted curve for stiffness and damping ratio.

**Figure 9 polymers-15-00257-f009:**
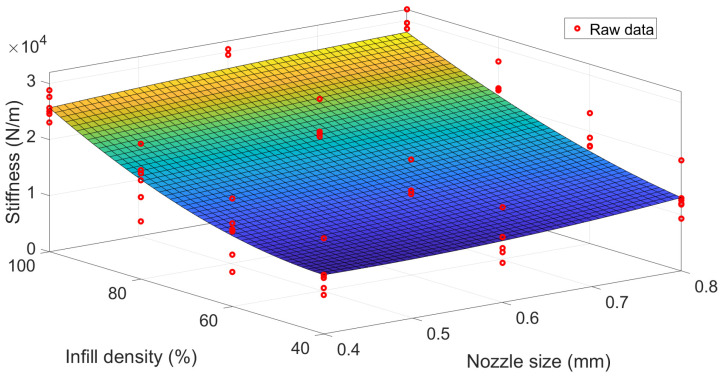
A nonlinear correlation between process parameter and stiffness.

**Figure 10 polymers-15-00257-f010:**
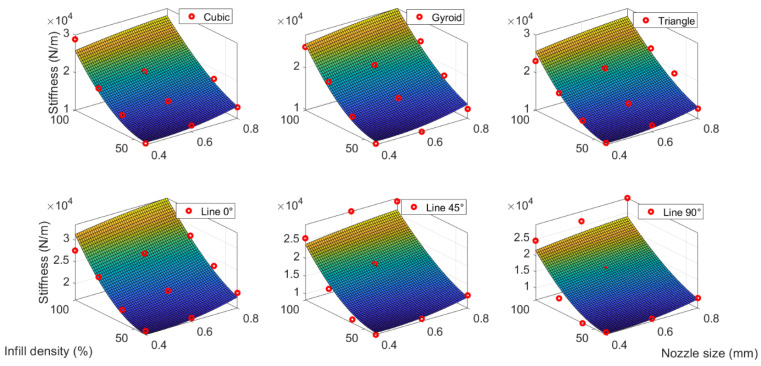
MLR correlation between process parameter and stiffness for different patterns.

**Figure 11 polymers-15-00257-f011:**
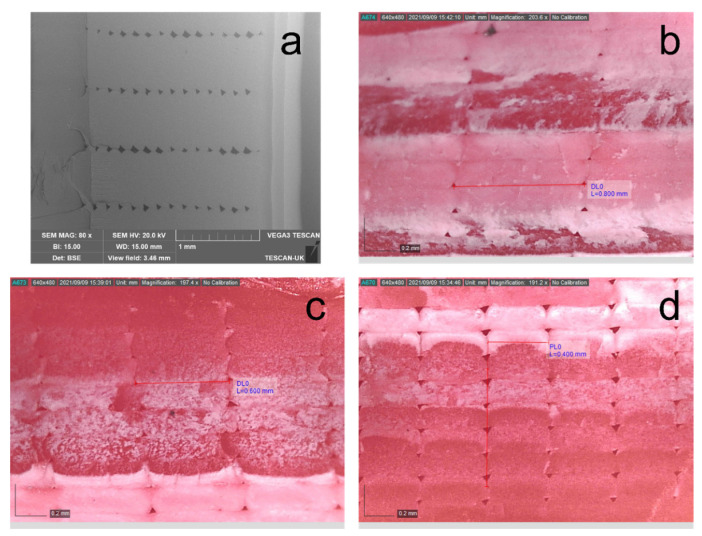
Cross-section of the specimen printed with 0° raster angle and 100% infill density. (**a**) Captured using SEM. (**b**) 0.8 mm nozzle size (**c**) 0.6 mm nozzle size (**d**) 0.4 mm nozzle size.

**Figure 12 polymers-15-00257-f012:**
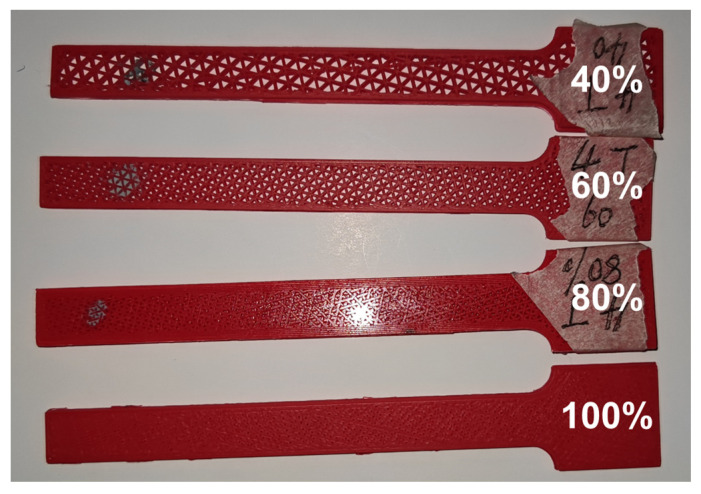
The specimens with different infill densities (0.4 mm nozzle size and triangle pattern).

**Table 1 polymers-15-00257-t001:** Some properties of FDM ABS filament.

Material	Ultimaker^®^ ABS Red
Tensile modulus	1618.5 MPa
Tensile stress at yield	39 MPa
Tensile stress at break	33.9 MPa
Flexural modulus	2070 MPa
Flexural strength	70.5 MPa
Melting temperature	225–245 °C

**Table 2 polymers-15-00257-t002:** ANOVA results (*n* represents the sample size).

3D Printing Parameters	Nozzle Size (mm)					Infill Density (%)						
	0.4 (*n* = 72)	0.6 (*n* = 72)		0.8 (*n* = 72)		40 (*n* = 54)		60 (*n* = 54)		80 (*n* = 54)		100 (*n* = 54)
Mean ξ	0.02495	0.01888		0.01817		0.025245		0.022951		0.020211		0.015674
*p*-value	8.716 × 10^−4^					1.350 × 10^−4^						
3D printing parameters	Infill pattern											
	Cubic (*n* = 36)		Gyroid (*n* = 36)		Triangle (*n* = 36)		Line 0° (*n* = 36)		Line 45° (*n* = 36)		Line 90° (*n* = 36)	
Mean ξ	0.016893		0.022756		0.018983		0.021163		0.022411		0.023914	
*p*-value	3.932 × 10^−10^											

**Table 3 polymers-15-00257-t003:** Correlation analysis results for nozzle size and infill density.

	Nozzle Size	Infill Density	Damping Ratio
Nozzle size	1		
Infill density	−3.60 × 10^−17^	1	
Modal damping ratio	−0.26971 (*p* = 5.93 × 10^−5^)	−0.40612 (*p* = 5.52 × 10^−10^)	1

**Table 4 polymers-15-00257-t004:** ANOVA results for process parameters and loss factor (*n* represents the sample size).

3D Printing Parameters	Nozzle Size (mm)					Infill Density (%)						
	0.4 (*n* = 24)	0.6 (*n* = 24)		0.8 (*n* = 24)		40 (*n* = 18)		60 (*n* = 18)		80 (*n* = 18)		100 (*n* = 18)
Mean η	0.0119	0.0118		0.0114		0.0118		0.0116		0.0118		0.0116
*p*-value	0.0741					0.2208						
3D printing parameters	Infill pattern											
	Cubic (*n* = 12)		Gyroid (*n* = 12)		Triangle (*n* = 12)		Line 0° (*n* = 12)		Line 45° (*n* = 12)		Line 90° (*n* = 12)	
Mean η	0.0115		0.0114		0.0118		0.0118		0.0117		0.0120	
*p*-value	0.0628											

**Table 5 polymers-15-00257-t005:** Correlation analysis results for nozzle size/infill density and loss factor.

	Nozzle Size	Infill Density	Loss Factor
Nozzle size	1		
Infill density	−6.757 × 10^−18^	1	
Loss factor	−0.2374 (*p* = 0.0446)	−0.0310 (*p* = 0.7960)	1

**Table 6 polymers-15-00257-t006:** Regression coefficients and p values for moderator analysis.

Regression Coefficient	Non-Standardised	Standardised	*p* Value
$p_{0}$	0.050	-	5.274 × 10^−13^
$p_{1}$	−0.029	−0.557	0.005
$p_{2}$	−0.000287	−0.742	0.001
$p_{3}$	0.000217	0.451	0.126

**Table 7 polymers-15-00257-t007:** Regression coefficients for Multiple Linear Regression (MLR) correlation between process parameters and stiffness.

Coefficients (with 95% Confidence Bounds)	Value
p00 (N/m)	3.353 × 10^4^
p10 (N/m/mm)	−5.698 × 10^4^
p01 (N/m)	−733.7
p20 (N/m/mm^2^)	3.892 × 10^4^
p11 (N/m/mm)	1136
p02 (N/m)	6.793
p21 (N/m/mm^2^)	−459.3
p12 (N/m/mm)	−4.251
p03 (N/m)	−6.931 × 10^−3^

**Table 8 polymers-15-00257-t008:** Regression coefficients for MLR correlation between process parameters and stiffness.

Pattern	$P_{00}$ Value	R-Square Value
Cubic	3.367 × 10^4^	0.8899
Gyroid	3.359 × 10^4^	0.9652
Triangle	3.346 × 10^4^	0.9260
Line 0°	3.920 × 10^4^	0.9012
Line 45°	3.178 × 10^4^	0.9397
Line 90°	2.945 × 10^4^	0.8848

**Table 9 polymers-15-00257-t009:** Some previous studies investigating the influence of the same process parameters influence on mechanical properties.

Study	Materials	Variable Parameters	Methods	Mechanical Properties
[11]	Polylactic acid (PLA)	Infill density	ASTM D638, ASTM D790, ASTM D256	Tensile strength, flexural strength and impact energy
[12]	PLA	Infill density, infill pattern	ASTM D638, ASTM D570, ASTM D695	Ultimate tensile strength, yield tensile strength, modulus of elasticity, elongation at break and toughness
[13]	Carbon fiber-reinforced PLA	Infill density, infill pattern	ASTM D638, ASTM D6110	Tensile strength and impact strength
[14]	ABS	Infill density, infill pattern	ASTM D638	Young’s modulus, yield strength and ultimate strength
[16]	PLA	Infill pattern	Compression test, fatigue test	Compression strength, fatigue strength
[17]	Polycarbonate (PC)	Infill pattern	ASTM D638, ASTM D5379	Tensile modulus, tensile strength, flexural modulus, bending strength, shear modulus, shear strength, charpy absorbed energy
[18]	PLA	Infill density, infill pattern	ASTM D638, ASTM D256, ASTM D695	Tensile strength, compression strength, impact strength
[20]	ABS	Nozzle size, infill pattern	Bending fatigue test	Fatigue life
[32]	thermoplastic polyurethanes (TPU)	Infill density, infill pattern	ISO 7743	Specific energy absorption, specific damping capacity
[33]	PLA	Infill pattern	Impact test for cantilever beam	Modal damping ratios

## Data Availability

The data presented in this study are available on request from the corresponding author.

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
