# Peer review of "Effect of 3D Printing Process Parameters on Damping Characteristic of Cantilever Beams Fabricated Using Material Extrusion"

_polymers, 2023, doi:10.3390/polym15020257_

Round 1
Reviewer 1 Report
The Manuscript describe about the effect of 3d printing process parameters. Below are my comments:
Concise the theoretical presentation in section 2.1
Geometrical figure1 should be presented nicely. The figure seems to be large.
Table 2 is just extra. You can mention this in text.
The modal damping ratio is of negative values for nozzle size and infill density. What is the reason?? Provide the explanation.
Minimize the theoretical justification for process parameter influence on damping behaviour section.
Provide the comparison table against the other proposed methods.
In overall, the manuscript is like a review paper. However, author have done extensive analysis with curve fitting technique. The coefficient values thus obtained need to be better justified with more explanation
Reviewer 2 Report
This work is based on the use of the mathematical model to reveal the dependencies of process parameters on the damping ratio of the sample. The topic is novel, but there are still some problems, mainly as follows:
Q1: The influence of printing on the damping property of materials should be involved in the abstract, and the importance and necessity of this study should be highlighted.
Q2: The nozzle diameter in Line 41 of the Introduction should be a variable in the first category of manufacturing parameters.
Q3: Can the mathematical model given in the article be applicable to the common thermoplastic materials used in FDM3D printing? Please specify.
Q4: What is the standard used in the model referred to in this paper?
Q5: The instruments and equipment used in the experiment should be written clearly, such as the 3D printer and the main printing parameters.
Q6: How do you choose the number of samples? Is there any basis?
Reviewer 3 Report
This work presents an empirical model of the relationship between process parameters and stiffness, and an attempt is made to establish quantitative interdependencies between process parameters, stiffness and modal damping ratios. The model better explains the effect of structural process parameters on the damping performance of FDM ABS cantilever beams. However, there are some major issues needed to be addressed before further consideration for publication in Polymers.
(1) It is suggested that the relationship between the damping coefficient c and the modal damping ratio be further indicated after Eq.(1) to make the later choice of modal damping ratio and loss factor clearer.
(2) In explaining the insignificant correlation between the process parameters and loss factor, would it be more convincing to mention the physical formula for the loss factor in structural damping theory?
(3) It is suggested that the full name (Multiple Linear Regression) be briefly introduced before the first mention of the MLR correlation (Table 8).
(4) Page 17, line 421, do nozzle size and raster angle have an effect on structural distributed mass and stiffness? Does infill density have an effect on the equivalent viscous damping coefficient?
(5) Is this model generalizable and can it explain the optimal infill density and pattern derived in the citation [32]?
Round 2
Reviewer 1 Report
Thank you for the modification performed in the manuscript as suggested. However, there should be comparison table which you need to compared with existing literatures. Its you responsibility to perform more study and come up with comparison table.
Author Response
Dear reviewer,
Thank you for your comment.
We have added Table 4 to compare our results with previous studies. However, as said in the reply before, we can only compare the trend instead of the exact values. In addition, only few previous studies investigated damping properties for 3D-printed structures.
Best regards,
Feiyang
Round 3
Reviewer 1 Report
The manuscript is not properly revised. The way author have mentioned tabular details are not convincing. I do suggested to revise manuscript again. The comparison table need to be placed before conclusion state.
You are not doing any detail investigation against the available methods. The aspect you have mentioned in the table is totally miss understandable. Its not good way to present comparison table in that way.
Do more research and study. You research seems to be not conducted correctly and be precise with your novelty.
Author Response
Dear Reviewer,
Thank you for your comments again.
We added one section before the Conclusion Section to summarise the related studies and tried making a comparison with our research.
However, as mentioned in the literature review part in the Introduction Section, most previous works investigated the influence of process parameters on tensile properties by tensile test as the method.
Only two previous studies focused on damping behaviours. Only one of them investigated the modal damping ratios by the impact test using a cantilever beam. Therefore, we can only compare our method and results with that one.
For the proposed method, compared with the previous study, our research applied the same methodology to measure the modal damping ratios as it is a standard process. But we tested different process parameters (infill pattern, infill density and nozzle size). The research also provided a theoretical justification which was never been discussed before. They are the novel parts of the manuscript.
For the comparison of the results, as we emphasised in the previous response, because the damping ratio is highly dependent on the geometry and dimension, we are only able to compare the trend between our and previous studies. Both studies investigated the raster angle influence but reached the opposite conclusion. And we have given theoretical support for the results we have observed in Section 3.4.
Kind regards,
Feiyang He
Round 4
Reviewer 1 Report
Thank you for the modification. The manuscript is now better. It will be better to upload the manuscript without changes mentioned. I mean the clear manuscript.